# A Generative Adversarial Network for Infrared and Visible Image Fusion Based on Semantic Segmentation

**DOI:** 10.3390/e23030376

**Published:** 2021-03-21

**Authors:** Jilei Hou, Dazhi Zhang, Wei Wu, Jiayi Ma, Huabing Zhou

**Affiliations:** 1College of Computer Science and Engineering, Wuhan Institute of Technology, Wuhan 430205, China; houjilei455@gmail.com (J.H.); whgcdxwuwei@163.com (W.W.); 2Research Institute of Nuclear Power Operation, Wuhan 430000, China; 3Electronic Information School, Wuhan University, Wuhan 430072, China; jyma2010@gmail.com

**Keywords:** image fusion, semantic segmentation, generative adversarial network, infrared image, visible image

## Abstract

This paper proposes a new generative adversarial network for infrared and visible image fusion based on semantic segmentation (SSGAN), which can consider not only the low-level features of infrared and visible images, but also the high-level semantic information. Source images can be divided into foregrounds and backgrounds by semantic masks. The generator with a dual-encoder-single-decoder framework is used to extract the feature of foregrounds and backgrounds by different encoder paths. Moreover, the discriminator’s input image is designed based on semantic segmentation, which is obtained by combining the foregrounds of the infrared images with the backgrounds of the visible images. Consequently, the prominence of thermal targets in the infrared images and texture details in the visible images can be preserved in the fused images simultaneously. Qualitative and quantitative experiments on publicly available datasets demonstrate that the proposed approach can significantly outperform the state-of-the-art methods.

## 1. Introduction

Image fusion is a significant technology to enhance images, aiming to synthesize images by integrating complementary information from several source images captured by different sensors [1,2]. It has been applied in many fields, including computer vision, medical image processing, and remote sensing [3]. Especially for object detection or tracking tasks, infrared and visible image fusion can highlight the object and offer more details. Infrared images can retain thermal radiation; thus, the thermal targets can be distinguished from their backgrounds based on thermal radiation differences, even in poor lighting conditions. Relatively, reflected light captured by visible light sensors could be represented in visible images; hence, visible images can provide texture details with high spatial resolution and definition, in a manner consistent with the human visual system [4,5]. Therefore, fused images can simultaneously preserve the texture details of visible images and the contrast of infrared images, benefiting subsequent tasks [6,7,8].

Over the past few years, increasingly, infrared and visible image fusion methods have been proposed. The methods can be classified as multi-scale transform-, sparse representation-, neural network-, subspace-, saliency-, and deep learning-based methods and hybrid methods. The existing fusion methods have three key points: image transformation, activity level measurement, and fusion rule designing. Based on these three points, a large number of successful fusion methods have emerged.

Although existing fusion methods can achieve good results, several aspects can be improved. First, artificial rule designing makes the hand-crafted methods more and more complex and time consuming [9]. Second, GAN-based infrared and visible image fusion lack a ground-truth; thus, it is difficult to design a comprehensive input for the discriminator to specify high-level tasks. There are two existing solutions to this problem, either taking visible images as references or learning a generative model via the dual-discriminator using both infrared and visible images as the input [10,11]. However, The first solution leads to thermal prominence, gradually reducing as the adversarial game proceeds. The dual-discriminator leads to a more complex network, and it is hard to keep the balance between the generator and dual-discriminator. Third, existing methods ignore the high-level semantic information of the source images. They carry a unified method to extract features from the source images, while source images are manifestations from the different aspects in different semantic regions. Therefore, their fusion results either partially lose the texture details of the visible image or the contrast of the infrared image. It is difficult for the whole image to retain the thermal radiation information of the infrared image and the texture details of the visible image at the same time.

To address these challenges, a generative adversarial network is proposed for infrared and visible image fusion based on semantic segmentation (SSGAN). Firstly, the proposed method is a deep learning-based method without artificial rule designing, which is an end-to-end fusion network with high efficiency. Secondly, the semantic information of the source image is taken into account. The source images are divided into the foregrounds and the backgrounds by utilizing the masks obtained by semantic segmentation. The foregrounds represent the salient semantic object, while the backgrounds represent the target with finer texture details. Thirdly, the generator is divided into the foreground- and the background-path to extract the corresponding information in source images, respectively. Finally, to overcome the lack of a ground-truth, the input images of the discriminator are designed by combining the foregrounds of the infrared images with the backgrounds of the visible images. This can promote the fusion images to preserve more texture details and the thermal radiation information concurrently. Qualitative and quantitative results reveal the advantages of the proposed SSGAN compared to other methods.

The contributions of the current work include the following aspects:

First of all, in order to extract the features of source images better, semantic segmentation is introduced into fusion tasks to divide the source images into the foregrounds and backgrounds. It is convenient to adopt a more targeted feature extraction strategy for the foregrounds and backgrounds.

Secondly, to extract various semantic information, a generator is designed with a dual-encoder-single-decoder model, encoding the corresponding features for different semantic regions by the dual-encoder and fusing different semantic features by the decoder.

Then, the image obtained by combining the foregrounds of the infrared images with the backgrounds of the visible ones is applied as the reference data to make the fused images retain abundant information.

Finally, in order to get the semantic information, the infrared and visible image fusion dataset is annotated, including 60 typical infrared and visible image pairs with labeled images. The dataset is available at https://github.com/Jilei-Hou/FusionDataset, accessed on 20 March 2021.

The rest of the paper is organized as follows. Section 2 briefly reviews the related research on infrared and visible image fusion, semantic segmentation, and generative adversarial networks. In Section 3, we describe the problem formulation, loss function, and network architecture of the proposed SSGAN. In Section 4, the proposed SSGAN is demonstrated for fusion on publicly available datasets (TNOand INO) with comparisons to five state-of-the-art approaches, followed by some concluding remarks in Section 5.

## 2. Related Work

In this section, the infrared and visible image fusion approaches, the recent research work related to semantic segmentation, as well as GANs and their variants are reviewed.

### 2.1. Infrared and Visible Image Fusion

Numerous infrared and visible image fusion methods have been proposed in recent years, which can be divided into seven categories, including multi-scale transform-based methods, sparse representation-based methods, neural network-based methods, subspace-based methods, saliency-based methods, deep learning-based methods, and hybrid methods.

Multi-scale transform-based methods [12,13] are the most familiar methods of infrared and visible image fusion, which are devoted to decomposing source images into several different scales, then fusing their different scales based on certain particular fusion rules. Finally, a final target fused image can be obtained according to corresponding inverse multi-scale transforms on the fused representations. The wavelet transform [14] is a typical multi-scale transform-based fusion method. Although it has achieved great success in infrared and visible image fusion, with the passage of time, some researchers have proposed more reliable fusion methods based on it. DTCWT [15] introduces a region-based fusion method to the dual-tree complex wavelet transform, which can generate fusion images with more thermal radiation information.

Sparse representation-based methods [16,17] have been successful used in infrared and visible image fusion, which can be divided into four steps. First, each source image is decomposed into several overlapping patches using a sliding window strategy. Second, an over-complete dictionary is learned from many high-quality natural images. According to the over-complete dictionary, the sparse representation coefficients are obtained by sparse coding for each patch. Third, the sparse representation coefficients are fused according to a given fusion rule. Finally, the learned over-complete dictionary and fused sparse representation coefficients are used to reconstruct the final fused image. The key of this method is to obtain the over complete dictionary. Wang et al. [18] classified image patches from source images into different groups based on morphological similarities to obtain sufficient information for sparse representation in dictionary construction.

Neural network-based methods have been popular ways of performing image fusion tasks over the past few years [19], due to the strong adaptability, fault tolerance, and anti-noise ability of neural networks. Kong et al. [20] presented a novel infrared and visible image fusion method based on a non-subsampled shearlet transform-spatial frequency-pulse coupled neural network, which can efficiently generate fused images with subjective visual performance.

Subspace-based methods [21,22] aim to project high-dimensional input images into a low-dimensional space or subspace, which helps to capture the intrinsic structure of the original images and remove the redundant information of natural images.

Saliency-based methods [23,24] imitate the feature of the human visual system, which is easily attracted to prominent objects, and improve the visual effect of fused images by preserving the integrity of salient objects. Zhang et al. [25] utilized salient target areas of an infrared image to determine the fusion weights, which can ensure that the fusion results have more significant thermal radiation targets.

Recently, since the rise of deep learning, a large number of deep learning-based fusion methods have been proposed [26,27]. Li et al. [28] decomposed infrared and visible images into base parts and detail content. Then, they fused the detail content by a deep learning network and fused the base parts in the traditional way, and finally, reconstructed the fused image. However, the entire fusion process is still like the traditional framework and not end-to-end. Later, they improved the entire fusion process by a deep learning framework with a dense block, named DenseFuse [29].

The above-mentioned infrared and visible image fusion methods all have their advantages and disadvantages. Hybrid methods [30,31,32] can combine their advantages to improve image fusion performance.

### 2.2. Semantic Segmentation

Semantic segmentation is a typical topic in computer vision that involves taking raw data (e.g., planar images) as the input and converting them into masks with highlighted areas of interest. Unlike other topics that only focus on the edge, semantic segmentation fully provides pixel-level image understanding in a human-perceived way by bringing together parts of an image that belong to the same target. Compared to other image-based tasks, semantic segmentation involves high-level cognition, which achieves the identification of the content and the corresponding location in the image.

Long et al. [33] proposed fully convolutional networks (FCNs), which replace the last fully connected output layer with a transposed convolution layer to obtain the prediction of each input pixel. UNet, proposed by Olaf et al. [34], is a typical encoder/decoder architecture that is a common network structure in semantic segmentation. The encoder uses a pooling layer to reduce the spatial dimension of input data gradually. Simultaneously, the decoder gradually recovers the target details and corresponding spatial dimensions through the deconvolution layer. There is usually a direct information connection between the encoder and decoder to better help the decoder recover its target details. Alex et al. [35] presented a deep learning framework for semantic segmentation named SegNet, the same as UNet, which has a typical encoder/decoder structure. To solve the problem of location information loss caused by multiple pooling, SegNet uses pooling with the index in max pooling, selects the maximum pixel, and records the pixel position in the feature map. In the anti-pooling process, the maximum value is restored to the original corresponding position according to the recorded coordinates, and zero padding for other positions. Google proposed a series of deeplab versions, which is a model used to deal with semantic segmentation. The latest version deeplabv3+ carries a deeper network, and all convolution layers and pooling layers are replaced by depthwise separable convolution [36]. Semantic segmentation has been successfully applied in other fields, especially image-to-image translation [37] and image alignment [38], but there is no application in fusion tasks. To the best of our knowledge, the current work is the first time that semantic segmentation has been introduced for addressing the image fusion task.

### 2.3. Generative Adversarial Networks

Goodfellow et al. [39] first proposed the concept of GAN, which has achieved impressive results in many fields, especially in image generation. The main inspiration of GAN comes from the idea of a zero-sum game in game theory. The GAN framework consists of two adversarial models: a generative model *G* and a discriminative model *D*. The generator can generate realistic images from random noise. The discriminator is trained to distinguish the generated data from the real data. It can make the generator learn the data distribution through a continuous game between the generative model *G* and discriminative model *D*. Compared with the traditional model, it has two different networks, not a single network, and the training method is adversarial training. The gradient update information of *G* in GAN comes from the discriminator *D*, not from data samples. The adversarial training between the generator and discriminator is continued until the discriminator cannot distinguish the generated samples. Subsequently, a relatively more realistic image sample can be produced by the generator.

The success that the original GAN has achieved is limited due to the unstable training process. To solve this problem, CGAN [40] extended GAN to a conditional model, where the generator and discriminator are conditioned on some extra information. DCGAN [41] slightly adjusts the network structure and uses the transposed convolution operation to generate clearer and more colorful images. InfoGAN [42] can learn more interpretable representations. WGAN [43] modifies the objective function of GANs to make the training process more stable.

GANs have achieved satisfactory results in infrared and visible image fusion. FusionGAN [10], proposed by Ma et al., was the first attempt to apply GANs to infrared and visible image fusion. It forces the fused image to obtain more texture details by introducing a discriminator with the visible image as real data. Subsequently, its variant [44] was proposed by introducing the target-enhancement loss to enhance the edge details of the fused image. However, as the adversarial game proceeds, the thermal prominence is gradually reduced. To address the problems, Ma et al. [11] introduced a GAN with a dual-discriminator using the infrared and visible image as input. The current work introduces a GAN based on semantic segmentation that extends the semantic-based image generation method to fusion tasks.

## 3. The Proposed Method

In this section, the proposed SSGAN for infrared and visible image fusion is presented. We begin with a formulation of the infrared and visible image fusion problem as image generation with GAN, and then, the loss function is discussed. Finally, the network architectures of the generator and the discriminator are described.

### 3.1. Problem Formulation

Infrared images can distinguish targets from their backgrounds due to the difference in thermal radiation even in poor lighting conditions. Visible images can represent texture details with a high spatial resolution. Meanwhile, both preserve the corresponding semantic information. Therefore, infrared and visible image fusion should keep both the radiation information of the infrared images and the texture details of the visible images. The fused images also need to reserve the high-level semantic information of both source images. Additionally, for different semantic targets, the information that fusion images want to retain from infrared and visible images is different. Specifically, the semantic object with thermal radiation information in infrared images takes precedence over the object’s information in visible images. In contrast, the semantic object with texture details in visible images takes precedence over the object’s information in infrared images.

In order to take semantic information into account in the fusion task, this paper proposes a generative adversarial network based on semantic segmentation, named SSGAN. According to the difference of thermal radiation information and texture details, each source image can be divided into a foreground and a background by the mask. The foreground is mainly composed of semantic targets that contain rich thermal radiation information in the infrared image, such as a person and a car. The background is mainly made up of semantic objects with rich texture details in the visible image. The proposed SSGAN can use different feature extraction methods for the foreground and background to better retain the information of the source image. The procedure of the proposed SSGAN is shown in Figure 1.

In the beginning, the mask with the infrared image’s semantic information is obtained by deeplabv3+, which is a successful model used for semantic segmentation. Then, we take the infrared image $I_{r}$, mask $I_{m}$, and visible image $I_{v}$ as the input of the generator *G*, whose output is a fused image $I_{f}$. However, It is hard to generate an informative fused image only by the generator. Thus, the discriminator *D* is employed to establish an adversarial game with the generator. Specifically, the generator aims to generate an image that can fool the discriminator. The discriminator is committed to distinguishing the generated image from the real image. As the adversarial game proceeds, a better fusion image can be generated by the generator. Moreover, in order to make the discriminator work better, we redesign the discriminator’s input based on semantic segmentation. This is obtained by blending the foreground of the infrared image and the background of the visible image. The production process of the real image $I_{d}$ can be formulated as in Equation (Equation 1).
(1)$$\begin{array}{c} I_{d}=I_{m}\circ I_{r}+\left(1-I_{m}\right)\circ I_{v}, \end{array}$$
where $I_{r}$, $I_{m}$, and $I_{v}$ represent the infrared image, mask, and visible image, respectively. ∘ means the Hadamard operator.

The ultimate goal of SSGAN is to learn a generator network *G* conditioned on a visible image $I_{v}$ and an infrared image $I_{r}$. The fused image $I_{f}$, generated by the generator *G*, is encouraged to be realistic and informative enough to fool the discriminator. By establishing an adversarial game between the generator *G* and discriminator *D*, the fused image $I_{f}$ will gradually contain more and more thermal radiation information of infrared image $I_{r}$ and the texture details of visible image $I_{v}$. Mathematically, the adversarial game between the generator *G* and discriminator *D* can be formulated as in Equation (Equation 2).
(2)$$\begin{array}{c} \min_{G}\max_{D}E\left[logD\left(I_{d}\right)\right]+E\left[log\left(1-D\left(I_{f}\right)\right)\right], \end{array}$$
where $I_{d}$ denotes the reference image that we want the generated image to be close to and $D(I_{d})$ denotes the classification results of the image $I_{d}$. $I_{f}$ denotes the fused image generated by the generator, and $D(I_{f})$ represents the classification results of generated data $I_{f}$. Through the adversarial process of the generator and discriminator, the divergence between the distribution of $I_{d}$ and $I_{f}$ will become smaller.

### 3.2. Loss Function

Due to the instability of the training process, the original GAN’s success was limited only by the game between the generator and discriminator. To alleviate this shortcoming, this paper introduces content loss $L_{con}$, which is tasked to constrain the similarity between the fused image and source images in the content. Moreover, structural similarity loss is adopted to ensure the structural similarity between the generated image and source images. Thus, the loss function of the generator *G* consists of three parts: adversarial loss $L_{adv}$, content loss $L_{con}$, and structural similarity loss $L_{SSIM}$. The whole loss function can be defined as in Equation (Equation 3).
(3)$$L_{G}=\lambda_{adv}L_{adv}+\lambda_{con}L_{con}+\lambda_{SSIM}L_{SSIM},$$
where $L_{G}$ denotes the total loss of the generator *G*, $\lambda_{adv}$, $\lambda_{con}$, and $\lambda_{SSIM}$ are used to control the trade-off, and the first item $L_{adv}$ means the typical adversarial loss between generator *G* and discriminator *D*, which is defined as in Equation (Equation 4).
(4)$$L_{adv}=\frac{1}{N}\sum_{n=1}^{N}{\left(D\left(I_{f}^{n}\right)-a\right)}^{2},$$
where *N* denotes the number of fused image patches, $I_{f}^{n}$ the *n*-th fused image patch, and *a* the value of whether the generator can fool the discriminator.

The second item $L_{con}$ represents the content loss. Each source image can be decomposed into thermal radiation information and the texture details in the infrared and visible image fusion task. Its pixel intensities characterize thermal radiation information, while its gradients can characterize texture details. Thus, we can gain an informative fused image by enforcing the fused image $I_{f}$ to have similar intensities and gradients to the source images. In a word, the content loss $L_{con}$ can be defined as in Equation (Equation 5).
(5)$$\begin{array}{cc} L_{con}=\frac{1}{HW} & (\mu_{1}∥I_{f}-I_{r}{∥}_{2}^{2}+\mu_{2}∥\nabla I_{f}-\nabla I_{r}{∥}_{2}^{2}\\ \; & +\mu_{3}∥I_{f}-I_{v}{∥}_{2}^{2}+\mu_{4}∥\nabla I_{f}-\nabla I_{v}{∥}_{2}^{2}), \end{array}$$
where *H* and *W* represent the height and width of the source images, respectively. ${∥\cdot ∥}_{2}$ stands for the two-norm, and ∇ means the gradient operator. The first two terms of $L_{con}$ aim to keep the thermal radiation information and texture details of infrared image $I_{r}$. the last two terms of $L_{con}$ aim to preserve the thermal radiation information and texture details contained in the visible image $I_{v}$. $\mu_{1}$, $\mu_{2}$, $\mu_{3}$, and $\mu_{4}$ control the trade-off.

It is worth noting that the texture details of visible images and the thermal radiation information of infrared images should be mainly preserved in the fused results, while the thermal radiation information of visible images and the texture detail of infrared images are of secondary importance. Therefore, $\mu_{1}>\mu_{3}$, $\mu_{2}<\mu_{4}$.

The last term $L_{SSIM}$ denotes the structural similarity loss of $L_{G}$, which can enforce the generated image and source images to have a similar structure. $L_{SSIM}$ can be formulated as in Equation (Equation 6).
(6)$$L_{SSIM}=\left(1-SSIM_{I_{f},I_{r}}\right)+\eta \left(1-SSIM_{I_{f},I_{v}}\right),$$
where $SSIM_{\left(\cdot \right)}$ denotes the structural similarity operation. It can be used to measure the structural similarity of two images, which models the loss and distortion according to the similarities in light, contrast, and structure information. η is used to control the trade-off. Mathematically, SSIM between images *x* and *y* can be defined as in Equation (Equation 7) [45].
(7)$$\begin{array}{c} SSIM_{x,y}=\sum_{x_{i},y_{i}}\frac{2\mu_{x_{i}}\mu_{y_{i}}+C_{1}}{\mu_{x_{i}}^{2}+\mu_{y_{i}}^{2}+C_{1}}\cdot \frac{2\sigma_{x_{i}}\sigma_{y_{i}}+C_{2}}{\sigma_{x_{i}}^{2}+\sigma_{y_{i}}^{2}+C_{2}}\cdot \frac{\sigma_{x_{i}y_{i}}+C_{3}}{\sigma_{x_{i}}\sigma_{y_{i}}+C_{3}}, \end{array}$$
where μ denotes the mean value, σ represents the standard deviation/covariance, and $C_{1}$, $C_{2}$, and $C_{3}$ are the parameters to make the metric stable.

To maintain the game between the generator *G* and the discriminator *D*, the discriminator is essential for distinguishing the real data and the generated data, the loss function of discriminator $L_{D}$ is defined as in Equation (Equation 8).
(8)$$\begin{array}{c} L_{D}=\frac{1}{N}\sum_{n=1}^{N}{\left(D\left(I_{d}\right)-b\right)}^{2}+\frac{1}{N}\sum_{n=1}^{N}{\left(D\left(I_{f}\right)-c\right)}^{2}, \end{array}$$
where *b* and *c* denote the value for whether the discriminator will trust and $D(I_{d})$ and $D(I_{f})$ represent the classification results of the real data and fused images, respectively.

### 3.3. Network Architecture

#### 3.3.1. Generator Architecture

As shown in Figure 2, the generator adopts a dual-encoder-single-decoder structure. It takes the infrared image $I_{r}$, visible image $I_{v}$, and mask $I_{m}$ as the input. Using the mask with semantic information, source images can be divided into the foreground and background. The feature characterization between foreground and background is distinct; therefore, two encoding paths are employed to extract their feature maps, respectively. Moreover, in the ideal foreground, the thermal radiation target is remarkable, and the texture details in the ideal background are rich. In order to achieve the effect as much as possible, two times the infrared image’s foregrounds and one time the visible image’s foreground are input to the foreground encoding path, while the background encoding path takes one time the infrared image’s background and two times the visible image’s backgrounds as the input. Finally, the feature maps of the foreground and background are concatenated as the decoder’s input to generate a fused image.

For each encoder path, four layers with 3 × 3 filters are adopted for feature extraction. Meanwhile, inspired by DenseNet [46], the dense connection is employed to realize feature reuse in each encoder path. Each layer’s output is cascaded as the input of the next layer. The decoder is a simple framework that contains four convolutional layers with 3 × 3 filters. We concat the outputs of the dual-encoder as the input of the decoder to reconstruct the fused image.

The specific settings of all layers are shown in Table 1. All kernel sizes were set as 3 × 3, and all strides were set as one with no pooling layers. To avoid exploding/vanishing gradients and speed up training and convergence, batch normalization (BN) and LReLUactivation function were applied.

#### 3.3.2. Discriminator Architecture

The discriminator plays an adversarial role against the generator to make the generator’s fused image more realistic and visual. Unlike the traditional single discriminator, it takes only one source image as real data or the dual-discriminator takes the infrared image and visible image as real data at the same time. The discriminator *D* of the proposed SSGAN is a simple and effective five-layer convolution neural network with well-designed real data. The discriminator architecture is shown in Figure 3. The front four layers are all 3 × 3 convolution layers with batch normalization (BN) and the LReLU activation function. In the last layer, the tanh activation function is employed to generate a scalar to estimate the input image’s probability from source images rather than generated by the generator *G*. The stride of all convolutional layers was set as two.

## 4. Experiments

This section focus on verifying the effectiveness of the proposed algorithm through a large number of qualitative and quantitative experiments on the TNO dataset (https://figshare.com/articles/TNOImageFusionDataset/1008029, accessed on 20 March 2021) and the INO dataset (https://www.ino.ca/en/technologies/video-analytics-dataset/, accessed on 20 March 2021). First, the experimental datasets and training details are introduced. Second, the parameters used in the proposed method are discussed. Third, a series of recent state-of-the-art methods for comparison and commonly used evaluation indicators are described in detail. Then, the experimental results on the TNO dataset and INO dataset are analyzed. Next, the time complexity is also considered. Finally, the ablation experiments related to the mask and discriminator are illustrated.

### 4.1. Dataset and Training Details

Dataset: To evaluate the proposed SSGAN, a set of experiments were produced. Different from existing methods, the proposed approach requires the masks with the infrared image’s semantic information. To obtain the masks, the model named deeplabv3+ needs to train on the infrared image dataset with the labeled image. Therefore, our datasets contain two parts: the dataset for semantic segmentation and the dataset for image fusion.

The dataset for semantic segmentation comes from the publicly available dataset for infrared and visible image fusion, which is well matched and does not require image registration [47,48,49,50]. We selected 136 pairs of infrared and visible images and labeled 60 infrared images to obtain 60 pieces of data that could be used to train deeplabv3+. However, 60 images were insufficient to train a satisfactory model. Therefore, the dataset was extended by flipping and mirroring to gain 180 images, of which 100 images were used for training and 80 images for verification. The remaining 76 images without labels were used for testing. Finally, we got 76 infrared images with the mask, which could be used to extract the foregrounds and backgrounds of source images.

The image fusion dataset was selected from these 76 pairs of infrared and visible images with the masks. Thirty-four infrared and visible image pairs with masks from the TNO dataset were selected to train our SSGAN. Through the mask with semantic information, training data can be divided into foreground and background regions. The expansion strategy of tailoring was adopted to obtain more training data; we set the stride as 14 and cropped each image into image patches with the size of 120 × 120. Eventually, the number of patch pairs for training was 10,168. The remaining 42 image pairs were used to test the performance of the proposed SSGAN.

Training details: In the training process, thirty-two pairs of infrared and visible image patches from the training data were selected to send to the generator. For the discriminator, thirty-two pairs of $I_{d}$ or fused image patches were used as the input. We first trained the discriminator *k* times with the optimizer solver Adam, then we trained the generator until reaching the maximum number of training iterations. In the testing process, we cropped the testing data without overlapping and input them into the generator *G* as a batch. Finally, the results of the generator were connected according to the sequence of cropping to get the final fused image.

### 4.2. Parameter Settings

The training parameters were crucial to make the network achieve the best fused results. The training parameters of SSGAN were set as follows: The batch size was 32, and the epoch was set as 20. The training step *k* of the discriminator was set to 2. The learning rate was set as 1 × 10${}^{-4}$ with exponentially decaying. Parameters a, b, and c are not specific numbers: a and b range from 0.7 to 1.2; c ranges from 0 to 0.3. For the other parameters: $\lambda_{adv}$, $\lambda_{con}$, $\lambda_{SSIM}$, $\mu_{1}$, $\mu_{2}$, $\mu_{3}$, $\mu_{4}$, and η, they were used to control the trade-off. $\mu_{1}=1.7$, $\mu_{2}$ = 4, $\mu_{3}=0.3$, $\mu_{4}=6$, $\lambda_{adv}=0.5$, $\lambda_{con}=1$, $\lambda_{SSIM}=0.5$, η=1.75.

### 4.3. Compared Methods and Objective Indexes

As is known to all, the superiority of a method often needs to be proven from qualitative and quantitative perspectives. For a qualitative comparison, the proposed SSGAN was compared to five state-of-the-art methods, including three traditional methods, i.e., GTF [51], DTCWT [15], and wavelet [14], and two deep learning-based methods, i.e., FusionGAN [10] and DenseFuse [29]. All these competitors were implemented based on publicly available codes, and the parameters were the default. It is worth noting that all deep learning-based methods ran on the same GPU GTX 1080Ti, while other traditional methods ran on the same CPU i5-6300HQ.

Although qualitative comparisons can evaluate the quality of fused images according to the human visual system, this will be affected by human subjective emotion. To evaluate the performance of fused images more comprehensively, quantitative experiments were also conducted based on the mathematical model’s evaluation index. Moreover, an objective evaluation index cannot fully reflect the quality of the fused images. Therefore, this paper uses four typical quantitative comparison indicators, namely entropy (EN), the standard deviation (SD), mutual information (MI), and visual information fidelity (VIF).

Entropy (EN) is a common method for statistical image features, reflecting the amount of information obtained from infrared and visible images. The mathematical definition of entropy is defined as in Equation (Equation 9) [52].
(9)$$EN=-\sum_{l}^{L-1}p_{i}\log p_{i},$$
where *L* denotes the gray level of the image and $p_{i}$ is the normalized histogram with the gray value of *i* in the fused image.

The standard deviation (SD) represents the dispersion of image gray value relative to the average gray value. The definition of the standard deviation can be defined as in Equation (Equation 10) [53].
(10)$$SD=\sqrt{\sum_{i=1}^{M}\sum_{j=1}^{N}{\left(F\left(i,j\right)-\mu \right)}^{2}},$$
where F(i,j) is the pixel value of the fused image *F* in the *i*-th row and the *j*-th column, the size of the fused image *F* is M×N, and μ is the average pixel value of the fused images.

Mutual information (MI) is a basic concept in information theory, which can measure the correlation between two random variables. In image fusion, mutual information is used to measure the correlation between the source images and the fused images. The definition of mutual information in infrared and visible image fusion is defined as in Equation (Equation 11) [54].
(11)$$MI=MI_{v,f}+MI_{r,f},$$
where $M_{r,f}$ denotes the correlation between infrared images and fused images and $M_{v,f}$ represents the correlation between visible images and fused images. Mutual information between any one source image *X* and fused image *F* can be defined as in Equation (Equation 12).
(12)$$MI_{X,F}=\sum_{x,f}p_{X,F}\left(x,f\right)\log\frac{p_{X,F}\left(x,f\right)}{p_{X}\left(x\right)p_{F}\left(f\right)},$$
where $p_{X}\left(x\right)$ and $P_{F}\left(f\right)$ represent the edge histograms of the source image *X* and the fused image *F*, respectively. $p_{X,F}\left(x,f\right)$ represents the joint histograms of the source image *X* and the fused image *F*.

The visual information fidelity (VIF) [55] metric measures the information fidelity of the fused images, which is consistent with the human visual system. VIF aims to build a model to compute the distortion between the fused images and source images.

### 4.4. Results on the TNO Dataset

#### 4.4.1. Qualitative Comparisons

Qualitative comparisons can evaluate the quality of the fusion image based on the human visual system. Since visible images conform to human visual habits, the fused results of the infrared and visible images should conform to human visual habits to a certain extent. To give some intuitive results on the fusion performance, we selected four typical image pairs from the TNO dataset for qualitative evaluation. The fusion results of the proposed SSGAN and the other five comparison methods are shown in Figure 4.

The first two columns in Figure 4 present the original infrared images and visible images. The third column is the semantic masks of the infrared images. The last column is the fusion results of the proposed SSGAN, and the remaining columns correspond to the fusion results of five comparison methods. From the results, we can find that all the methods can obtain satisfactory performance in preserving texture details and thermal radiation information, which can help maintain useful information as much as possible.

However, different fusion methods have their own unique performance, and the infrared thermal target is very obvious in the fused results of GTF and FusionGAN. In other words, the thermal radiation information of the infrared images is preserved successfully. On the other hand, although the salient target information in the fused results obtained by GTF and FusionGAN is well preserved, some of the texture details in visible images are missing. In contrast, for other comparison methods like DTCWT, wavelet, and DenseFuse, the targets (e.g., the human or the building) in the fused images are not as obvious as those in the infrared images. This means that the thermal radiation information in the infrared images is not well preserved, but their texture details are well preserved.

In general, all the comparison methods exploit the information in the source images, either thermal radiation information or texture details. Different from all the comparison methods, the proposed SSGAN is devoted to preserving both thermal radiation information and texture details at the same time. For example, in the second column, the fused images of SSGAN highlight people while retaining the texture of the ground. A similar phenomenon can also be observed in other examples. This demonstrates that the proposed SSGAN has better performance than the other state-of-the-art methods in terms of simultaneously preserving thermal radiation information and texture detail information.

#### 4.4.2. Quantitative Comparisons

The proposed SSGAN was further compared with the five state-of-the-art methods quantitatively on 26 image pairs. The results are summarized in Figure 5. The proposed SSGAN can generate the largest average values on the four metrics. It is worth mentioning that the larger the EN, the more information is contained in the fused images. Thus, the largest EN means that the proposed SSGAN preserves the most information compared to the other methods. The largest SD demonstrates that the fused images of SSGAN have the highest contrast between the thermal targets and their backgrounds. The largest MI means that considerable information is transferred from source images to the fused image, which indicates a good fusion performance. The largest VIF means that the fused results of SSGAN are more consistent with the human visual system. These results demonstrate that the proposed method can reserve the most information and have the largest image contrast; meanwhile, the results of the proposed method are consistent with the human visual system.

### 4.5. Results on the INO Dataset

The proposed method and the other five comparison methods were further evaluated on the INO dataset, which is also a dataset commonly used for infrared and visible image fusion. Sixteen visible and infrared image pairs were selected from the video named Treesandrunner for both qualitative and quantitative comparisons.

#### 4.5.1. Qualitative Comparisons

The qualitative fused results of the proposed SSGAN and the other five comparison methods are shown in Figure 6. We can see that the infrared image contains rich thermal radiation information, which helps distinguish the target from the background. In contrast, the visible image preserves rich texture details, which is convenient for human observation. Considering the fusion results, the texture information can be well retained by all six methods, but SSGAN is more like a visible image in overall brightness. Moreover, compared with the other methods, the proposed method has obvious advantages in retaining the thermal radiation information, which can be proven by the fact that the people in the image are better highlighted.

#### 4.5.2. Quantitative Comparisons

In addition to qualitative experiments, the proposed SSGAN was further compared with the above-mentioned competitors quantitatively on 16 image pairs from the INO dataset with four metrics, i.e., entropy (EN), standard deviation (SD), mutual information (MI), and visual information fidelity (VIF). The results are summarized in Figure 7. We can find that the proposed SSGAN produced the largest averages on all four metrics, and SSGAN had obvious advantages with respect to the MI and VIF metrics. For the EN and SD metrics, only GTF was equivalent to SSGAN in some fused results. These results prove the superiority of the proposed method in quantitative indexes.

The runtime comparison of six methods is also provided in Table 2. From the results, the conclusion can be drawn that the proposed SSGAN can achieve comparable efficiency compared with the other five methods.

### 4.6. Ablation Experiments

#### 4.6.1. Experiment Related to the Mask

In order to generate a high quality fused image and overcome the shortcoming of lacking of a ground-truth, we designed the input image $I_{d}$ of the discriminator based on semantic segmentation. The masks were used to divide the infrared and visible images into foregrounds and backgrounds, respectively. Then, we added the foregrounds of the infrared image and backgrounds of the visible one to get $I_{d}$. Since the generation of $I_{d}$ was closely related to the mask, the role of $I_{d}$ was verified, and the benefit of the mask was also verified to a certain extent. Therefore, we compared the fused results of SSGAN with the results by replacing $I_{d}$ with infrared image $I_{r}$ and visible image $I_{v}$, respectively.

The comparison results are given in Figure 8. We can see that the proposed SSGAN preserved more background details than replacing $I_{d}$ with the infrared image $I_{r}$. In the first row, the fused result of SSGAN keep the texture of the tree better. In the second row, the sky’s color is more realistic, and the texture on the ground is clearer. Compared to the results of replacing $I_{d}$ with the visible image $I_{v}$, SSGAN has higher contrast in the foreground. In both two rows, the proposed SSGAN can highlight persons better, which is conducive to target detection. These results demonstrate that $I_{d}$, which is designed based on the mask, does improve the quality of the fused image. Therefore, the conclusion can be drawn that the mask obtained by semantic segmentation is indeed useful for fusion tasks.

#### 4.6.2. Experiment without the Discriminator

To verify the effect of the game between the generator and discriminator in GAN, we removed the discriminator, and the whole network merely contained the generator. Thus, the adversarial relationship no longer existed. The training goal was to minimize the content loss $L_{con}$ and SSIMloss $L_{SSIM}$. As shown in Figure 9, compared to the fused results using the network without discriminator, the results of SSGAN have a clear advantage in that it retains more texture details and the results more consistent with the human visual system.

## 5. Discussion and Conclusions

This paper presents a new approach for infrared and visible image fusion. The key characteristic of the proposed method is that it applies semantic segmentation to image fusion, and the network structure adopts a dual-encoder-single-decoder to extract features. The proposed method uses the powerful GAN technique, and the reference data of the discriminator are redesigned based on semantic segmentation. The experiments on the TNO and INO datasets demonstrate that the proposed approach yields results that consistently outperform the state-of-the-art methods, both qualitatively and quantitatively.

However, the current work verifies the effectiveness of semantic segmentation for the fusion task, and the segmentation network is trained separately. In the future, we will try to use a unified network for fusion and segmentation and build segmentation consistency loss between the fused image and source images. The segmentation consistency loss can constrain the semantic consistency between the fused image and the source images, which will enforce the network to generate a fused image with clearer semantic targets and richer texture details. 

## Figures and Tables

**Figure 1 entropy-23-00376-f001:**
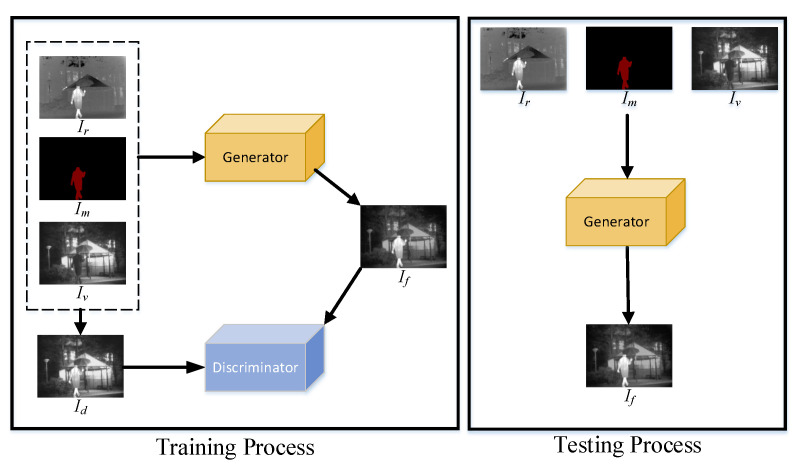
The entire procedure of semantic segmentation GAN (SSGAN) for image fusion. $I_{r}$, $I_{m}$, $I_{v}$, and $I_{f}$ denote the infrared image, mask, visible image, and fused image, respectively. $I_{d}$ is used as the reference data to input into the discriminator, and its generation process is described later.

**Figure 2 entropy-23-00376-f002:**
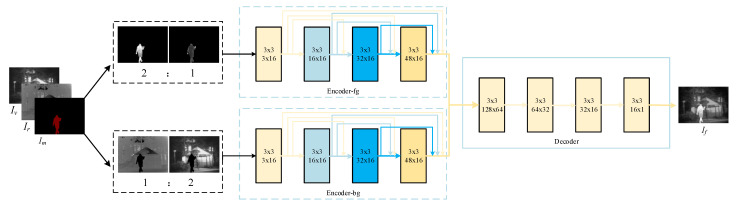
The overall architecture of the generator. $I_{r}$, $I_{m}$, $I_{v}$, and $I_{f}$ denote the infrared image, mask, visible image, and fused image, respectively. Encoder-fg and Encoder-bg are used to extract the features of the foreground and background, respectively, and the decoder is used to reconstruct the fused image.

**Figure 3 entropy-23-00376-f003:**
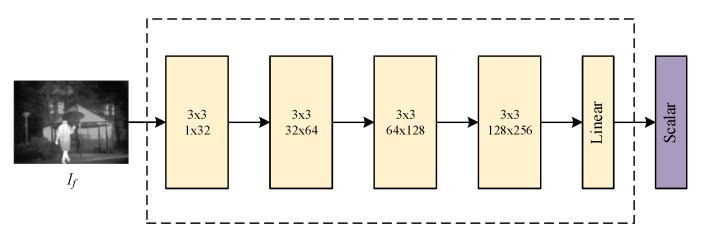
The overall architecture of the discriminator.

**Figure 4 entropy-23-00376-f004:**
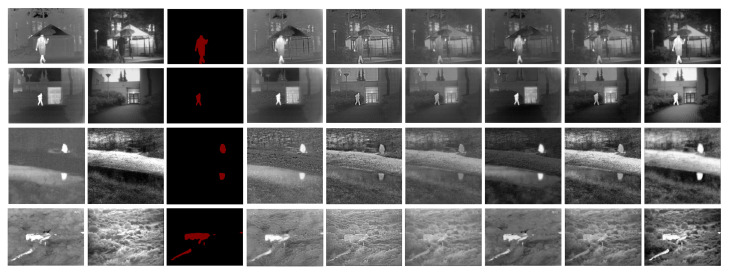
Qualitative fusion results on four typical infrared and visible image pairs. From left to right: infrared images, visible images, mask, results of GTF, DTCWT, wavelet, FusionGAN, DenseFuse, and the proposed SSGAN.

**Figure 5 entropy-23-00376-f005:**
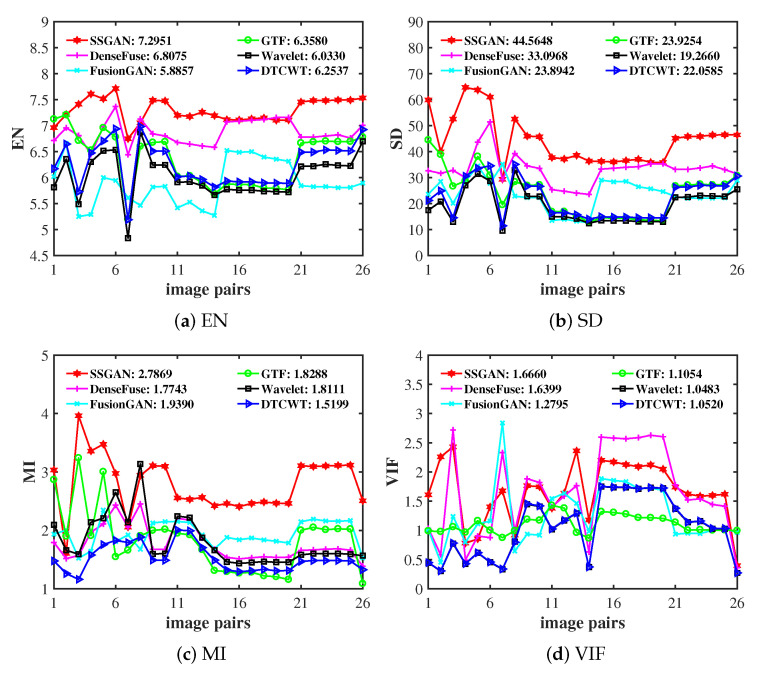
Quantitative comparisons of the four metrics, i.e., entropy (EN), SD, MI, and visual information fidelity (VIF), on the TNOdataset. The five state-of-the-art methods, DTCWT, GTF, wavelet, DenseFuse, and FusionGAN, are used for comparison.

**Figure 6 entropy-23-00376-f006:**
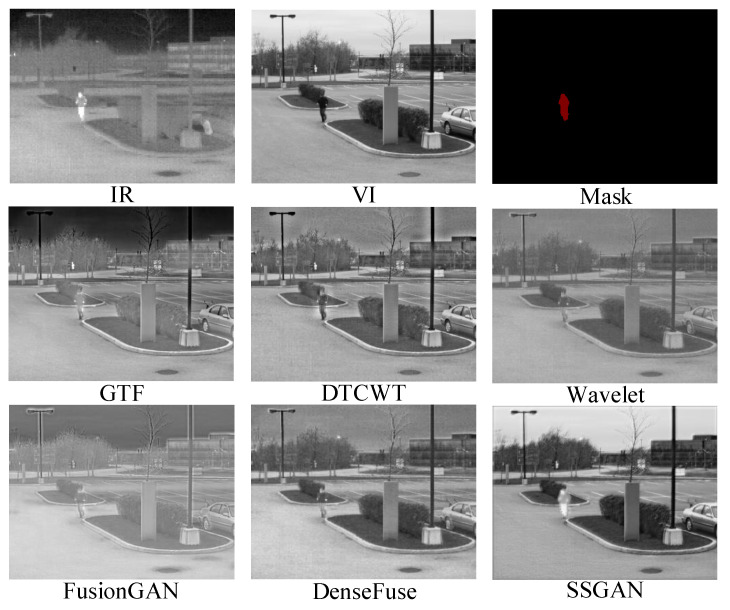
Qualitative fusion results on the INOdataset. The first row is the infrared image, visible image, and mask. The rest two rows (from left to right, top to bottom) are the fusion results of GTF, DTCWT, wavelet, FusionGAN, DenseFuse, and SSGAN.

**Figure 7 entropy-23-00376-f007:**
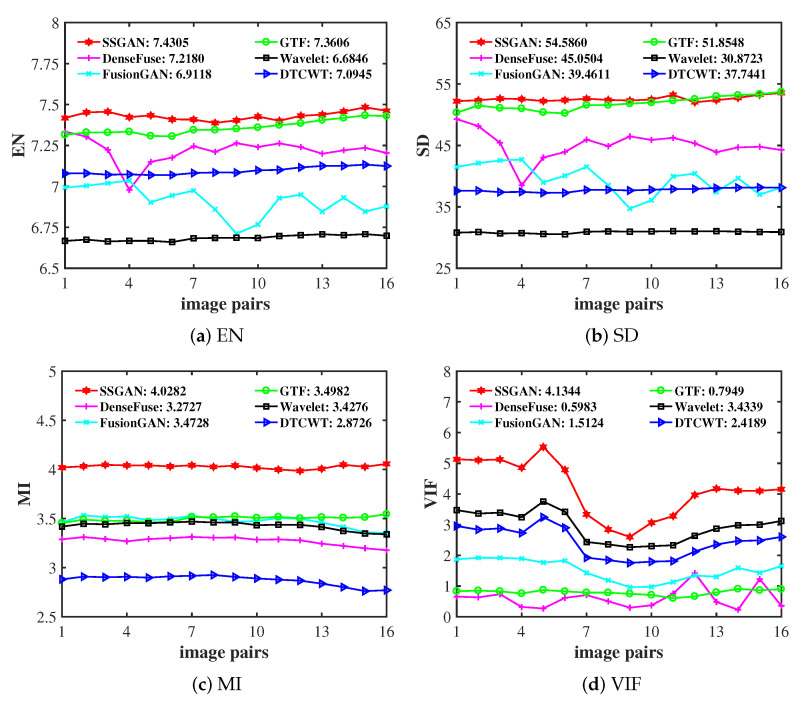
Quantitative comparisons of the four metrics, i.e., EN, SD, MI, and VIF, on the INO dataset. The five state-of-the-art methods DTCWT, GTF, wavelet, DenseFuse and FusionGAN are used for comparison.

**Figure 8 entropy-23-00376-f008:**
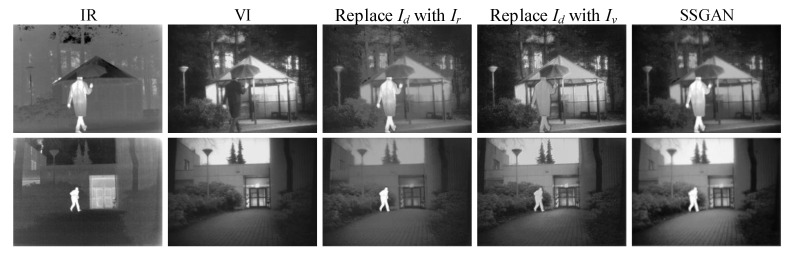
Fused results of the experiment related to the mask. From left to right: infrared images, visible images, results of replacing $I_{d}$ with $I_{r}$, results of replacing $I_{d}$ with $I_{v}$, and results of the proposed SSGAN.

**Figure 9 entropy-23-00376-f009:**
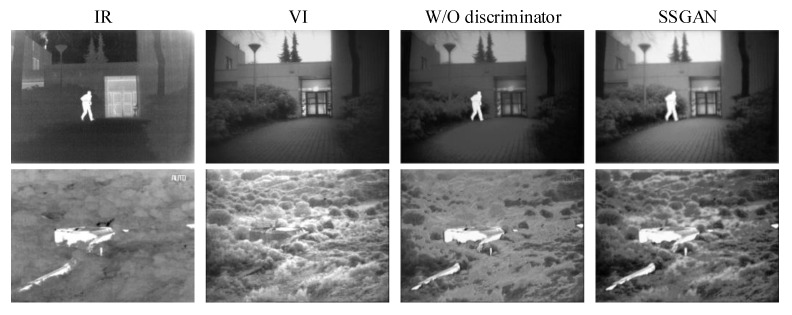
Fused results of the experiment related to the discriminator. From left to right: infrared images, visible images, results of the SSGAN without the discriminator, and results of the proposed SSGAN.

**Table 1 entropy-23-00376-t001:** Input/output channels of the generator’s convolutional layers.

	Layers	Input Channel	Output Channel	Activation
Encoder-fg	conv1	3	16	LReLU
	conv2	16	16	LReLU
	conv3	32	16	LReLU
	conv4	48	16	LReLU
Encoder-bg	conv1	3	16	LReLU
	conv2	16	16	LReLU
	conv3	32	16	LReLU
	conv4	48	16	LReLU
Decoder	conv1	128	64	LReLU
	conv2	64	32	LReLU
	conv3	32	16	LReLU
	conv4	16	1	Tanh

**Table 2 entropy-23-00376-t002:** Runtime comparison of the six methods on the TNO dataset and the INO dataset. Each value denotes the mean of the runtimes of a certain method on a dataset (unit: seconds).

Dataset	GTF	DTCWT	Wavelet	FusionGAN	DenseFuse	SSGAN
TNO	2.3495	0.2193	0.2121	0.0606	0.1538	0.1070
INO	0.6954	0.1139	0.1589	0.0803	0.1625	0.0886

## Data Availability

The dataset is available at https://github.com/Jilei-Hou/FusionDataset, accessed on 20 March 2021.

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
