# Peer review of "A Generative Adversarial Network for Infrared and Visible Image Fusion Based on Semantic Segmentation"

_entropy, 2021, doi:10.3390/e23030376_

Round 1
Reviewer 1 Report
This paper presents a generative adversarial network for infrared and visible image fusion based on semantic segmentation (SSGAN). The authors introduce semantic segmentation to improve the performance of existing fusion methods. Contributions include the following four aspects. First, semantic segmentation is adopted to guide feature extraction. Second, the authors design a generator with the dual-encoder-single-decoder model, encoding the corresponding features for different semantic regions by the dual-encoder and fusing different semantic features by the decoder. Third, a new reference data I_d is designed based on semantic segmentation as the discriminator's input. Finally, the authors release an infrared and visible image fusion dataset, including 60 typical infrared and visible image pairs with labeled images. A set of experimental results on publicly available datasets compared to the state-of-the-arts demonstrates the superiority of the proposed strategy.
My overall impression of this paper is that it is well-written and pretty comprehensive. The work seems interesting, and the technical contributions are solid. The paper needs minor revisions to get accepted, and my concerns are presented in the following.
(1) In Page 6, L229, the authors mention that “...c is the value that the generator can fool the discriminator”, but c does not appear in Eq. (4). Should c be replaced by a? A similar problem occurs in Eq. (8).
(2) In Figure 2, the input arrow in front of the Encoder-fg and Encoder-bg is not connected to the encoder's first layer, which is ambiguous.
(3) In Page 7, L251-253, the authors mention that “...to extract the desired features better, the inputs of two encoder paths are both determined to be 1: 2.”, the authors are suggested to explain this in detail.
Author Response
We would like to thank the reviewer and the associate editor for their time and constructive comments, which enable us to greatly improve the quality of our manuscript. We also provide a marked-up copy of our manuscript that highlights changes made to the original version. Next we provide point-by-point responses to the review comments.
(1) In Page 6, L229, the authors mention that “...c is the value that the generator can fool the discriminator”, but c does not appear in Eq. (4). Should c be replaced by a? A similar problem occurs in Eq. (8).
Reply:
Thank you for pointing out this issue. We have fixed it in our new version.
(2) In Figure 2, the input arrow in front of the Encoder-fg and Encoder-bg is not connected to the encoder's first layer, which is ambiguous.
Reply:
Thanks. We have modified Figure 2 to make sure it is clear.
(3) In Page 7, L251-253, the authors mention that “...to extract the desired features better, the inputs of two encoder paths are both determined to be 1: 2.”, the authors are suggested to explain this in detail.
Reply:
Thank the reviewer for this valuable suggestion. We have explained this problem in detail in our revision. The feature characterization between foreground and background is distinct, therefore, two encoding paths are employed to extract the feature maps of them, respectively. Moreover, in the ideal foreground, the thermal radiation target is remarkable, and the texture details of the ideal background are rich. In order to achieve the effect as far as possible, two times the infrared image foregrounds and one time the visible image foreground are input to the foreground encoding path, while the background encoding path takes one time the infrared image background and two times the visible image backgrounds as input. Finally, the feature maps of the foreground and background are concated as the decoder's input to generate a fused image.

Reviewer 2 Report
This paper utilized the semantic segmentation (SSGAN) method for infrared and visible image fusion. The proposed framework has been tested with a variety of datasets and has given promising results.
- There is a crucial issue related to problem formulation, and the presentation of the results was very confused. The author(s) needs to define more details on research gaps and the adapted research methodology approach.
-Highlight the drawbacks associated with the proposed model and providing useful, practical tips.
- Add literature of review, and the results need to be contrasted and compared with the results of the other state-of-the-art methods in the literature to support the main findings thoroughly.
- Enhance the abstract to focus only on the objectives, methodology, and add quantitative results.
The results and discussion section did not arrange well and did not give any value to the reader.
It needs in-depth discussions and explanations of equations and their benefits. Also, it needs more comprehensive evaluations and comparisons with other researchers (mention in the related work) to validate the obtained results supported by graphical and tabular data.
- Avoid using many references together like in L97, L102 , L114 , etc.
The research paper should be written in the third person's perspective; words such as "we", "our," etc., need to be avoided.
-Enhance the resolution of figures such as Fig.1, 2, etc.
- Enhance the formatting of the paper, such as text after L212, L239, etc.
-All equations which are not derived by the authors should be cited (provide a reference).
-The text has too long sentences, which makes the meaning unclear. Consider breaking this into multiple sentences, such in page 5 (L41-L43; L43-L47, etc.).
-The language used should adequately inform the reader, and Proofreading is mandatory for English grammar and style.
- Many grammatical or spelling errors that make the meaning unclear and sentence construction errors, punctuation errors. The following some examples:
L27: Therefore, fused images can preserve the texture details of visible images and contrast of infrared images simultaneously, benefiting subsequent tasks [5,6].
… Should be ….
Therefore, fused images can simultaneously preserve the texture details of visible images and contrast of infrared images, benefiting subsequent tasks [5,6].
L42 : dual-discriminator lead to … Should be …. dual-discriminator leads to
L43 : complex network and it is hard to keep … Should be …. complex network. It is hard to keep…..
L45 : source image, they carry a unified … Should be …. source image. They carry a unified…..
L221: formulated as follow: … Should be …. formulated as in equation 2. (check all other equations for the same).
Author Response
We would like to thank the reviewer and the associate editor for their time and constructive comments, which enable us to greatly improve the quality of our manuscript. We also provide a marked-up copy of our manuscript that highlights changes made to the original version. Next we provide point-by-point responses to the review comments.
-There is a crucial issue related to problem formulation, and the presentation of the results was very confused. The author(s) needs to define more details on research gaps and the adapted research methodology approach.
Reply:
Thank the reviewer for this valuable suggestion. In our new revision, we formulate infrared and visible image fusion as a problem that needs to preserve both low-level features and high-level semantic features. The low-level features are mainly the contrast of the infrared image and the texture details of the visible image. Existing fusion methods only consider the low-level features and ignore the high-level semantic features. The proposed method attempts to take the high-level semantic features into account.
In our revision, we have made the following revisions. Firstly, we formulate infrared and visible image fusion as a problem that needs to preserve both low-level features and high-level semantic features (the first paragraph in section 3.1). Secondly, we define the categories of semantic objects and the reasons for such classification (the second paragraph in section 3.1). Next, we introduce the details of the proposed method (the third paragraph in section 3.1). Finally, we formulate the fusion problem into an objective function based on GAN (the last paragraph in section 3.1).
-Highlight the drawbacks associated with the proposed model and providing useful, practical tips.
Reply:
We have discussed the drawbacks associated with the proposed model and provided a feasible scheme in the discussion and conclusion section. Our current work just verifies the effectiveness of semantic segmentation for the fusion task, and the segmentation network is trained separately. In the future, we will try to carry a unified network for fusion and segmentation, and build segmentation consistency loss between the fused image and source images. The segmentation consistency loss can constrain the semantic consistency between the fused image and the source images, which will enforce the network to generate a fused image with clearer semantic targets and richer texture details.
-Add literature of review, and the results need to be contrasted and compared with the results of the other state-of-the-art methods in the literature to support the main findings thoroughly.
Reply:
Thanks for your suggestion. We have added literature of review in related work. Further, we have compared the proposed method with the other state-of-the-art methods, such as GTF [1], DTCWT [2], Wavelet [3], FusionGAN [4], and DenseFuse [5]. The comparison results are shown in the experiment section, which demonstrate that the proposed method yields results that consistently outperform the state-of-the-art methods both qualitatively and quantitatively.
[1] Ma, J.; Chen, C.; Li, C.; Huang, J. Infrared and visible image fusion via gradient transfer and total variation minimization. Information Fusion 2016, 31, 100–109.
[2] Lewis, J.J.; O’ Callaghan, R.J.; Nikolov, S.G.; Bull, D.R.; Canagarajah, N. Pixel-and region-based image fusion with complex wavelets. Information fusion 2007, 8, 119–130.
[3] Chipman, L.J.; Orr, T.M.; Graham, L.N. Wavelets and image fusion. Proceedings of the International Conference on Image Processing, 1995, pp. 248–251.
[4] Ma, J.; Yu, W.; Liang, P.; Li, C.; Jiang, J. FusionGAN: A generative adversarial network for infrared and visible image fusion. Information Fusion 2019, 48, 11–26.
[5] Li, H.; Wu, X.J. DenseFuse: A fusion approach to infrared and visible images. IEEE Transactions on Image Processing 2018, 28, 2614–2623.
-Enhance the abstract to focus only on the objectives, methodology, and add quantitative results.
The results and discussion section did not arrange well and did not give any value to the reader. It needs in-depth discussions and explanations of equations and their benefits. Also, it needs more comprehensive evaluations and comparisons with other researchers (mention in the related work) to validate the obtained results supported by graphical and tabular data.
Reply:
Thanks for pointing out this issue. We have rewritten the abstract. Please track the changes in our marked-up copy.
In the experimental section, the proposed method has been compared with other existing methods on the public datasets. From the qualitative and quantitative results, we can see that the proposed method has unique advantages. Moreover, in the discussion and conclusion section, we describe the limitation of the proposed method and provide an effective improvement scheme.
-Avoid using many references together like in L97, L102, L114, etc.
The research paper should be written in the third person's perspective; words such as "we", "our," etc., need to be avoided.
Reply:
In our revision, we have fixed this issue to avoid using many references together. Moreover, we have taken the suggestion and recent paper into consideration and used the third person's perspective to revise our paper where appropriate.
-Enhance the resolution of figures such as Fig.1, 2, etc.
Reply:
We have enhanced the resolution of figures in our revision.
-Enhance the formatting of the paper, such as text after L212, L239, etc.
Reply:
We have enhanced the formatting of the paper in our revision as suggested.
-All equations which are not derived by the authors should be cited (provide a reference).
Reply:
Thank the reviewer for this valuable suggestion. We have provided references to all equations which are not derived by us in our revision, such as equations (9)-(11).
-The text has too long sentences, which makes the meaning unclear. Consider breaking this into multiple sentences, such in page 5 (L41-L43; L43-L47, etc.).
Reply:
Thanks, we have fixed this issue to avoid having too long sentences in our revision.
-The language used should adequately inform the reader, and Proofreading is mandatory for English grammar and style.
Reply:
We have carefully revised the English grammar and style to ensure that readers can easily read our paper.
-Many grammatical or spelling errors that make the meaning unclear and sentence construction errors, punctuation errors. The following some examples:
L27: Therefore, fused images can preserve the texture details of visible images and contrast of infrared images simultaneously, benefiting subsequent tasks [5,6].
… Should be ….
Therefore, fused images can simultaneously preserve the texture details of visible images and contrast of infrared images, benefiting subsequent tasks [5,6].
L42: dual-discriminator lead to … Should be …. dual-discriminator leads to
L43: complex network and it is hard to keep … Should be …. complex network. It is hard to keep…..
L45: source image, they carry a unified … Should be …. source image. They carry a unified…..
L221: formulated as follow: … Should be …. formulated as in equation 2. (check all other equations for the same).
Reply:
We have fixed these above issues in our new version. Moreover, we have read the full paper to check the grammatical or spelling errors.

Round 2
Reviewer 2 Report
The manuscript has been enhanced to an acceptable level that it could be published in the current form.